# Generative Mechanisms for Scientific Knowledge Transfer in the Food Industry

**Karla Zimpel-Leal** [1,*] and **Fiona Lettice** [2]

1   CIRCLE, Faculty of Social Sciences, The University of Sheffield, Sheffield S10 2TN, UK
2   Norwich Business School, University of East Anglia, Norwich NR4 7TJ, UK; fiona.lettice@uea.ac.uk
*   Correspondence: k.zimpel-leal@sheffield.ac.uk

**Abstract:** This paper investigates the generative mechanisms for scientific knowledge transfer in the food industry, addressing the sustainability of knowledge transfer projects related to health, safety and regulation. Different levels of analysis examine structure, agency and interactions within a multilevel framework. The main research questions are: (1) what are the key generative mechanisms within science–industry knowledge transfer? and (2) what are the implications of these mechanisms to policy? This research applies explaining-outcome process-tracing by investigating different knowledge transfer projects, utilising empirical data from 52 in-depth interviews with food scientists and food SMEs, 17 supporting documents and 16 observations. Systematic combining is used to develop a narrative from empirical data, where the evidence leads to the formation of the most plausible explanation. This is followed by the abstraction of mechanisms which are then matched to a suitable theoretical framework. The results from the study show a range of predominant mechanisms that drove scientific knowledge transfer including nonpecuniary incentives, reputation, opportunity, instrumental rationality, self-interest, strategic calculation, aggregation, learning and adaptive self-regulation. The overall conclusion is that the construction of relationships based around social norms, autonomy and relatedness are more dominant than those focused on financial incentives or transaction cost theories.

**Keywords:** scientific knowledge transfer; generative mechanisms; food sector; mechanismic explanation

## 1. Introduction

Science–industry knowledge transfer has long been considered important to explain innovation in the food industry, however, it has also highlighted challenges raised between actors [1]. Knowledge transfer has been found to improve growth trajectories through the network settings promoted by knowledge transfer projects [2], nonetheless trust and language are identified as key obstacles that hamper the transfer of scientific insights to firms in the food sector [3]. Scholars and practitioners posit a strong focus on economic gains that firms can potentially attain from knowledge transfer activities with either science-academic partners or other firms [4]. Mechanisms such as economic contracts [5], social exchange [6], autonomy [7], engaged scholarship [8], and learning [9] are seen as useful tools that can improve knowledge transfer and increase competitive advantage.

A recognition of the importance of science to industry knowledge transfer in this context led to the motivation to investigate the generative mechanisms for science–industry knowledge transfer in the food industry, especially to Small and Medium Sized Enterprises (SMEs) in the UK. The food industry in the UK it is composed of over 6000 SMEs which are not always able to directly access the latest innovation and technology. Whilst there is a plethora of studies investigating the motivations for firms to engage in knowledge transfer with scientists, there is a paucity of inductive studies that look for fine-grained explanations of why this process happens from science to industry. Science to industry knowledge transfer is increasingly important for universities and other research organisations to

demonstrate the value and relevance of their research. Whereas there is evidence that scientists are increasingly engaging in commercial activities [10], what remains unknown is whether or not they are driven by the same for-profit mechanism as in the traditional private sector. Thus, the main research questions addressing this gap in the food industry are: (1) what are the key generative mechanisms within science–industry knowledge transfer? and (2) what are (if any) the implications of these mechanisms to policy?

To understand the generative mechanisms of science–industry knowledge transfer, this study proposes an inventive analytical framework, at both macro and micro levels (structure, agency and interaction). This framework builds on previous studies that focus on either macro level relationships (i.e., [11–13]) or micro level relationships (i.e., [14,15]). This process enables us to move from a limited understanding of knowledge transfer as a singular functional perspective to a cognitive understanding of knowledge transfer as situated in organisational structures, individual motivations and their interactions. Furthermore, the explanation is based on a grounded approach that outlines the dominant mechanisms that cause the transfer of new scientific insights to SMEs, using different theoretical lenses for each mechanism.

We apply this framework to three collaborative knowledge transfer projects in the food industry. These were specifically selected to examine how knowledge is transferred between the largest food research institute in the UK and food manufacturer SMEs, via in-depth interviews, observations and document analysis. Drawing on our analysis, we offer the conceptualisation of three dominant generative mechanisms for each level of abstraction. The macro or situational mechanisms include nonpecuniary incentives, reputation and opportunity whilst the micro or action-formation mechanisms include instrumental rationality, self-interest and strategic calculation. The interactive or transformation mechanisms are aggregation, learning and adaptive self-regulation. These insights contribute to a deeper understanding of scientists' environmental constraints and opportunities to collaborate with industry and their individual motivations that are seemed to be beyond economics and extrinsic rewards to include nonpecuniary, social and personal aspects related to intrinsic motivation. We conclude with the implications that this mechanismic perspective of knowledge transfer has for policies.

## 2. Literature Review

The literature identifies interdependent streams of science–industry knowledge transfer, and it includes those focused on motivations and assessing the psychological side of relationships; those focused on the process itself, roles, its structure and various stages; those focused on the economical and performance impact of knowledge transfer; and those focused on relational aspects such as trust, learning, networks and social exchange.

From a motivational perspective, academic capitalism [16] and entrepreneurial universities [17] describe academics as promoters of commercialisation, emphasising the for-profit motive of the entrepreneurial scientist. Whilst there is evidence that scientists are increasingly engaging in commercial activities, such as patenting, spin-off company formations [15] and licensing [18], what remains unknown is whether they are driven by the same for-profit mechanism as in the private sector. Other motivators for academic–industry knowledge transfer have been around the reputation of scientific peers and the availability of institutional technology transfer support [19].

Citations, prizes and other similar forms of peer recognition have typically been recognised as the predominant extrinsic rewards for academic career advancement or increased salaries and as the main motivators to engage with industry. However, D'Este and Perkmann's [15] study on the entrepreneurial university and on the motivations for academic–industry engagement, concludes that most academics engage with industry to further their research rather than to commercialise their knowledge. Similarly, Göktepe-Hulten and Mahagaonkar's [20] and Lam's [21] studies of the motivations for scientists to pursue commercial activities report that reputation is a stronger incentive than financial rewards. Furthermore, Iorio et al. [22] suggest that the "mission" is a key motivation to

pursue knowledge transfer activities, where the academic scientist advances the societal role of universities.

Studies have also proposed that individuals invest time, energy and effort into knowledge transfer to create collaborative networks and cooperative relationships [12]. Although a variety of motivations for academics to engage with industry have been identified, studies agree that there are both intrinsic and extrinsic motivations. Intrinsic motivations include reputation and research support, whilst extrinsic commercialisation-maximising motivations include increased income.

There is a recognition that knowledge transfer is a process with various stages of transfer and the factors that correlate to the difficulty of transfer activities. For example, Böcher [11] looks at scientific knowledge transfer as the connection between research (R), integration (I), and utilisation (U), the RIU model. Within this RIU-model, scientific knowledge is produced in the science system (research), and science-based problem solutions are utilised within practice by political actors (utilisation). The key mechanism in this model is integration, which is the step that connects the science sphere to utilisation. The Triple Helix model [23] also goes beyond dyadic relationships and offers insights into the dynamics of the relationship among research, industry and government. One limitation of these models is that they focus at the macro level activity and tend not to include an individual level of analysis.

Studies which propose that academics are motivated by monetary profit, suggest that researchers use patents to increase their income, and pursue relationships with firms to access equipment or exploit other research-related opportunities [24]. Taking a transaction cost economics perspective, Katz and Martin [5] suggest that academic–industry collaborations can be prolonged by economic commitments, which create a 'locked-in condition' between partners, therefore ensuring that the cooperation is continued and endured. Another economic lens used to view knowledge transfer is the prisoner's dilemma of collective action (game theory) which suggests that information asymmetry and independent strategies within firms that are transferring knowledge can cause conflicting interests in learning, which could lead to the end of the collaboration [9].

Engaged scholarship also affects knowledge transfer relationships and knowledge is more likely to be adopted when the stakeholders have been involved in the process of knowledge creation [8]. It is important that collaborative work between research and practice produces knowledge that is more penetrating and insightful than when researchers work alone. A perception misalignment between SME entrepreneurs and academics can hinder innovation and lead to the failure of many knowledge transfer initiatives [25]. This perspective offers a traditional view of science–industry knowledge transfer activities, with economic-type gains to businesses and reputational-type gains to researchers.

From a relational perspective, Adler and Kwon [26] work on social capital and suggest that informal social ties are superior conduits for knowledge sharing. Furthermore, the social capital dimensions of networks—structure, cognition and relation—affect the transfer of knowledge. Inkpen and Tsang [27] examine how organisations acquire knowledge depending on their positions within networks and conclude that organisations should build and use their social capital proactively for efficient knowledge transfer. This view is also shared by Yli-Renko et al. [28], who suggest that knowledge transfer is facilitated by the intensive social interactions of various actors.

Another widely cited theory that explains knowledge transfer is absorptive capacity [29], which implies that knowledge transfer is only successful if the receiver of the information has prior related knowledge in order to recognise the value of what they are receiving and to be able to assimilate it effectively. Thus, knowledge stickiness is a major barrier to knowledge transfer when the recipient lacks absorptive capacity, which affects the execution and implementation of the transfer.

Social exchange theory refers to situations where rewards or punishments are provided in recurring interactions. Muthusamy and White [6] found that social exchanges such as reciprocal commitment, trust and mutual influence are positively related to knowledge

transfer. Whereas economists assume that firms' behaviours towards knowledge transfer are motivated by self-interest, social exchange theorists believe that knowledge transfer can be motivated by a broad array of interests and that self-interest and group interests can coexist [30]. The knowledge flow between scientific networks and industry, and its relation to human resource issues, identifies two perspectives on knowledge transfer: cognition and competencies versus careers and incentives [31].

Liyanage et al. [32] propose a process model of knowledge transfer using the theory of communication and theory of translation. They argue that knowledge transfer is facilitated by collaboration (theory of communication) and transformation of knowledge into a usable form (theory of translation). This concurs with Holden and Von Kortzfleisch [33] who argue that the perceived utility of knowledge from the receiver determines the effective translation and quality of the knowledge transferred. They used translation theory as an applicable analogy to explore the nature of knowledge transfer and go a step further to explain that the process is only successful if the source understands their own knowledge and if they understand what it means to the receiver. Thus, this translation involves the interpretation of the same knowledge in a different manner or context in order to be accessible and absorbed.

SMEs working with university research centres tend to rely on relational trust and self-interest [34]. Studies have considered knowledge transfer from an individual's trust perspective and the importance of boundary spanning individuals to build trust with other organisations [35]. Both interpersonal and interorganisational trust are considered key drivers for knowledge exchange performance, with the former affecting institutionalisation, and the latter associated with lowered costs of negotiation and conflict.

Autonomy and trust are considered important mechanisms for knowledge transfer [7]. Where individual autonomy is impractical, organisations can minimise the effects of low autonomy by fostering institutional and interpersonal (benevolence-based) trust. The mechanisms of trust have either assumed a self-interest angle [34], a cooperative competency perspective [13] or a relational trust angle [8]. These studies investigate how trust, both interpersonal and interorganisational, affects knowledge transfer and consequently organisational performance. However, these studies do not explain the generative mechanisms of trust.

Overall, there are four broad perspectives of mechanisms. The first has a strong focus on motivations such as rewards, peer recognition, enhanced reputation and access to knowledge and resources. The second type of mechanism focusses on economic commitments to ensure collaboration and increased competitive advantage. The third offers an emphasis on the process of knowledge transfer whilst the fourth type emphasises a relational and social capital dimension. Most of these studies position these mechanisms from either a macro or micro perspective but never from both angles, nor do they consider interaction mechanisms.

This study inductively investigates the generative mechanisms for scientific knowledge transfer, which are broadly related to the four types of mechanisms observed in the current literature. It applies an analytical framework that looks at structural mechanisms by looking at rules, norms and resources; agency mechanisms by looking at motivations and beliefs; interaction mechanisms by looking at actions and interactions. This analysis differs from current literature as each mechanism is inductively reasoned from rich data whilst previous studies have concluded their mechanisms from a predetermined correlation with other variables.

Despite the voluminous literature on knowledge transfer, studies have not yet investigated generative mechanisms. So far it has been difficult to provide in-depth insight as to why knowledge transfer occurs. This study aims to advance prior research by offering an alternative perspective, with alternative ontological and epistemological assumptions about cause and effect, grounding the knowledge transfer literature in a robust framework based on a process-oriented view. Through an inductive study with scientists and food

sector SMEs, this study investigates the generative mechanisms for knowledge transfer and whether or not they have an impact on this type of relationship.

## 3. Materials and Methods

The methodology used in this study was process-tracing [36], with abductive reasoning [37] as the main analytical method. Process-tracing is a within-case method of analysis and a key technique for capturing the presence or absence of generative mechanisms [36]. It goes beyond the identification of correlations between independent variables and outcomes, with the ambition to trace underlying generative mechanisms that involve interpretation, contextualisation and abstraction by analytically or temporally ordering the empirical data.

This study applied explaining-outcome process-tracing to build a theoretical explanation from the empirical evidence. The goal was to trace the generative mechanisms that explain the knowledge that is transferred from science to industry in the food sector. By investigating different knowledge transfer projects and utilising a multiple framework, empirical data from 52 in-depth interviews with scientists and directors of food SMEs, 17 supporting project documents, and 16 observations were reviewed and analysed. Systematic combining through abductive reasoning was then used as the analytical method to abstract the generative mechanisms and match corresponding theoretical explanations. Abductive reasoning develops a narrative from empirical data, where the evidence leads to the formation of the most plausible explanation, followed by the abstraction of mechanisms which are then matched to a suitable theoretical framework. Process-tracing is well placed to move theory beyond either/or debates to empirical applications in which both agents and structures matter. It moves us away from correlational arguments and as-if styles of reasoning toward theories that capture and explain the world as it really works. Process-tracing also offers the ability to make connections between different theories.

Mechanisms are analytical constructs that draw useful connections between social instances [38]. Generative mechanisms are unobservable; we do not observe causality but make inferences about it, hypotheses about them generate observable and testable implications. Mechanisms cannot establish causality but they allow explanatory accounts by first utilising historical or causal narratives and then abstracting the mechanisms. Hedstrom and Ylikoski [39] (p. 51), defined mechanisms as "consisting of entities (with their properties) and the activities that these entities engage in, either by themselves or in concert with other entities. These activities bring about change, and the type of change brought about depends on the properties of the entities and how the entities are organised spatially and temporally."

A mechanismic explanation advocates that there is no mechanism that operates solely at the macro level. In other words, there are no macro-level entities that possess the capacity to act or the capability of producing outcomes, hence the importance of looking at individual actions. However, that is not to say that macro-level explanations are not important. They are very relevant to establish correlations between macro-variables and are a useful shorthand, however they need further explanation at the micro-level. A mechanismic explanation takes the position that a macro phenomenon such as knowledge transfer in a science–industry setting must ultimately be grounded in explanatory mechanisms that involve individual actions and interactions.

Mechanism-based explanations aim to provide a plausible account of the generative mechanisms that are necessary to explain how, under certain contextual conditions, an observed phenomenon has emerged. This perspective aims to identify the generative mechanisms that allow us to explain with some confidence "how" and "why" something happened rather than merely observing that something happened [40].

There were three projects that were analysed in depth. These were collaborative projects that occurred between the largest publicly funded food research institute in the UK and food manufacturer SMEs. Although the SMEs involved had first-hand access to the projects' findings, there were no royalties involved, and once the projects were completed,

findings could be accessed by other food manufacturers. The projects were selected from rigorous selection criteria that included purpose, variety, evidence, industry presence and accessibility, as summarised in Table 1.

**Table 1.** Selection criteria.

|  |  | **BACCHUS** | **SUSSLE** | **NIS** |
|---|---|---|---|---|
| (1) | Purpose | To transfer knowledge related to bioactives and peptides from Research Institute science base to food manufacturers. | To transfer knowledge related to assessment of food poisoning bacteria *Clostridium Botulinum* in chilled foods, consequently extending shelf-live and reducing waste. | To transfer knowledge to food producers related to the nutritional composition and labelling of their foods. |
| (2) | Variety | Food and health. | Food safety. | Food regulation. |
| (3) | Evidence | Food manufacturers were able to use findings from the project to back their products' health claims. | Food manufacturers were able to significantly reduce their waste by increasing shelf-live and reducing *C. Botulinum* levels. | Small and medium food producers were able to comply to new nutritional labelling regulation by accessing an affordable and efficient service. |
| (4) | Presence of industry | This project had a direct impact on 16 food manufacturers SMEs and several food manufactures were indirectly benefited. | This project had a direct impact on three food manufacturers SMEs and over 200 food manufactures were indirectly benefited. | This project had an impact on over 250 food manufacturers SMEs. |
| (5) | Accessibility | Project leader agreed to be part of the study; interviews were established with Research Institute scientists and food manufacturers involved in the project. | Project leader agreed to be part of the study; interviews were established with Research Institute scientists and food manufacturers involved in the project. | Project leader agreed to be part of the study; interviews were established with Research Institute scientists and food producers involved in the project. |

From a pool of 11 projects, the chosen ones met the criteria more closely. The goal was to trace the generative mechanisms that explain knowledge that transferred from this institute to food SMEs. One project was health driven (BACCHUS), one safety driven (SUSSLE) and one regulation driven (NIS), as summarised in Table 2.

**Table 2.** Project descriptions.

| Project | Driver | Description |
|---|---|---|
| BACCHUS | Food and Health | The main objective of BACCHUS was to understand the cardiovascular benefits from food bioactives, which are compounds found naturally in many different fruits and vegetables. The aim was to develop tools and resources that could be used by food manufacturers to support their health claims. |
| SUSSLE | Food Safety and Food Waste | The aim of the SUSSLE project was to understand the levels of a bacterium *Clostridium botulinum* in raw food ingredients to help the food industry deliver safe chilled foods more sustainably, by extending shelf-life and reducing waste. |
| NIS | Food Regulation | The aim of NIS was to provide a cost-effective nutritional labelling service to food SMEs by offering calculations based on IFR's Food Databanks. A new regulation states that all food producers must provide a detailed nutritional label. |

Primary data for the research were gathered through two rounds of semi-structured interviews, each lasting between 40 and 90 min, with key stakeholders from the research institute involved in the projects. Participants from food SMEs were also interviewed to confirm that knowledge was transferred and utilised within their organisations. Stakeholders outside the project that were involved in the wider environment, including government, policy and funding bodies, were also interviewed, as summarised in Table 3. The interview

data were triangulated between interviews with different participants, and the various documents and the observations examined. In total, there were 52 semi-structured interviews that formed the primary dataset.

**Table 3.** Interviews per participant group.

| Participant Position | Organisation | Number of Interviews |
|---|---|---|
| Senior Researcher | Food Research Institute | 12 |
| Research Leader | Food Research Institute | 23 |
| CEO or Director | SMEs | 14 |
| Head or Senior Manager | Government, policy and funding organisations | 3 |

Documents from a wide range of sources were used as evidence. These included official documents from the food research institute such as project contracts and terms of agreements. Other documents from mass-media outputs such as magazines, newspapers, internet resources and archived documents were also accessed. Data sources such as official websites, background documents and publicly available reports, interviews and articles were also used to extend the findings. The documents added context, and gave further information that could be utilised during the face-to-face interviews, provided contact detail information, and confirmation that the industry and government are utilising the knowledge from the research institute, including their results and findings. The documents also gave an indication of the scope of reach that their projects had, not only to the immediate stakeholders involved but also to the wider industry.

Observations were conducted in 16 events, including networking events, workshops, seminars, and sector-specific conferences. The purpose of attending these events was twofold. Firstly, the events provided current information regarding challenges affecting the food industry in general. Secondly, they provided a platform to network with various actors and established contacts that were later used for interviews or obtaining access to documents and reports. They also provided an opportunity to discuss issues around innovation and knowledge transfer with the most prominent figures in the sector. Observations focused on problem-solving discussions and talks, where the interaction and opinions of different actors could be captured. They attempted to record information about (a) what are the challenges occurring in the food industry during the period of this study, including backgrounds, processes and outcomes; (b) how are solutions proposed for current challenges; (c) which processes facilitate and inhibit innovation and knowledge transfer; (d) what are the characteristics of different actors and an understanding of power dynamics (e.g., funders $\times$ research institute $\times$ food SMEs; complex or simple, ambiguous $\times$ clear).

NVivo 12 software was used to assist in data management in terms of classification and organisation and subsequent qualitative content analysis. The evidence was organised into macro codes for each level of the framework, as shown in Table 4:

**Table 4.** Macro codes.

| Macro Codes for Structure Level | Macro Codes for Agency Level | Macro Codes for Interaction Level |
|---|---|---|
| • Rules<br>• Norms<br>• Resources | • Motivations<br>• Beliefs | • Interactions<br>• Actions |

Identifying the generative mechanisms for knowledge transfer on each project was an iterative process of constant matching of what was found, the broader context, theoretical constructs found in the literature, and the emerging contextualised explanation. Dubois and Gadde [37] explain this process as systematic combining, a process based on abductive

reasoning. The process of systematic combining leads to directing and redirecting the search for more sources of information, and possible explanatory theories to reconstruct the most acceptable causal explanation [37,41]. The cornerstone of systematic combining is 'matching' which means "going back and forth between framework, data sources and analysis" [37] (p. 556), This process differs from the mainstream positivist literature where the researcher begins from propositions [42] or a 'tight and prestructured' framework [43] or follows specific steps from 'getting started' to 'reaching closure' [44]. An abductive approach seeks the most plausible explanation among several alternatives. Abductively derived explanations require support from deductively (theory) and inductively (empirical) sourced evidence. Thus, it can be problematic due to the subjective nature of choosing between plausible alternatives.

Another limitation of abductive reasoning is that it presupposes the existence of theoretical frameworks that can explain the suggested generative mechanisms. These mechanisms, although firmly based in generally accepted theories, could only be inferred, but not tested. The theoretical frameworks are supposed to guide the researcher in their approach, as in the analysis. One limitation is that either those theoretical frameworks are lacking, or they are ill-suited, leaving the researcher vulnerable to biases or forced to use an ill-adapted theory. When a theory does exist, it is often insufficiently specified and rarely tailored to the problem at hand. In this study, although an engagement in theory through systematic combining was a significant contribution, there were parts of theories that were used to explain the various mechanisms, and not a single theory was found to be all-encompassing.

The type of mechanism was classified into different levels: situational, action-formation and transformational, as explained in the analytical framework in Figure 1. The choice of the predominant mechanisms for each project involved abductive reasoning, where the evidence led to the formation of the most plausible explanation. For example, when it became clear that projects were affected by the UK research impact agenda at the structure level, it was necessary to look for further evidence as to why scientists responded to this structural constraint in different ways.

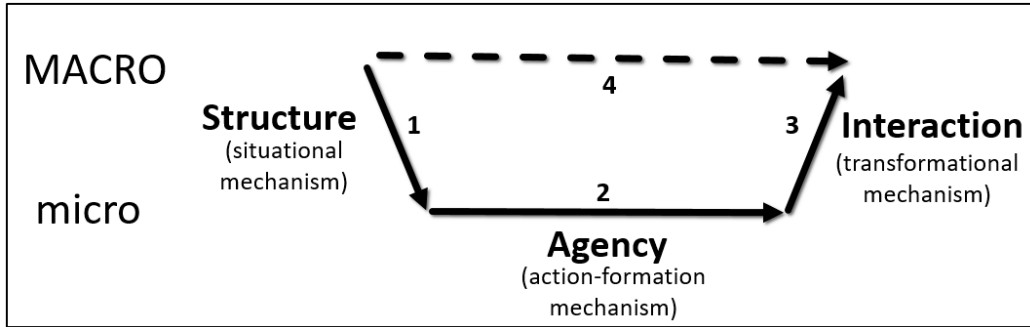

**Figure 1.** Analytical framework.

The next analytical stage is to link the generative mechanisms from the empirical findings to the extant body of theory to find a suitable conceptual framework, through the iterative process of systematic combining [37]. A close examination of macro and micro theories allowed for the explanation of the various mechanisms. For the situational mechanisms, there was an investigation into macro theories that explain structural constraints and opportunities, whilst for the transformational mechanisms there was an investigation into macro theories that explain individuals' interactions such as decision-making type of theories. The action-formation mechanisms were explained by micro theories that ranged from rational to more behavioural types. These theories aided the most suitable explanation for each mechanism and included, among others, the macro theories of compliance and trust, and the micro theories of rational choice and the theory of reasoned action.

The multilevel nature of the model encouraged more rigorous thinking about how certain theories might apply to multiple levels of analysis and about the potential boundary conditions of a mechanism approach. This framework is based on the macro–micro–macro model of social action proposed by sociologist James Coleman and is referred to as Coleman's boat (or bathtub) [45]. Coleman's framework has been widely used in sociology to explain social interactions. Transposed to an organisational setting, this framework can be substantially informative regarding the micro foundations of the phenomenon under study.

The analytical framework comprises of three main levels: structure, agency and interaction, as shown in Figure 1. The structure level unveils the situational mechanism related to the structural side of the project. Elements of the system including norms, rules and resources help to identify the structural constraints and opportunity for action. The agency level unveils the action-formation mechanism related to agency and the explanation of the actors' behaviours and choices on the basis of their motivations and beliefs. The actors are the research leaders and scientists from the research institute and also individuals from the food SMEs involved in each project. The third analytical level is the interaction, where the transformational mechanism is abstracted, relating to the explanation of an outcome which unfolds over time, on the basis of the interaction and actions of different individuals.

This analytical framework promotes going beyond analysing relationships between phenomena exclusively on the macro level (arrow 4). It identifies the "situational mechanisms by which social structures constrain individuals' action and cultural environments shape their desires and beliefs (arrow 1), describes the action-formation mechanisms linking individuals' desires, beliefs, etc., to their actions (arrow 2), and specifies the transformational mechanisms by which individuals, through their actions and interactions, generate various intended and unintended social outcomes (arrow 3)" [39] (p. 58).

Whereas correlational analysis involves identifying antecedents regularly conjoined with outcomes, mechanismic analysis consists of specifying the 'mechanism' that underlies and generates empirical regularities or outcomes. These mechanisms explain why a phenomenon happened, and knowledge of their operation allows results to go beyond correlations or the relationship between variables.

## 4. Results

The following section will present the results from the analysis of the empirical data to identify the underlying generative mechanisms that are driving scientific knowledge transfer from a food research institute to food manufacturer SMEs.

### 4.1. Situational Mechanisms

The data analysis showed three predominant situational mechanisms: nonpecuniary incentives, reputation and opportunity. These mechanisms are related to the structural side of the project, including norms, rules and resources that shows the structural constraints and opportunities, in other words, the macro influence on more micro behaviour.

### 4.1.1. Nonpecuniary Incentives

Nonpecuniary incentives were important in shaping individual actions and beliefs. Changes in the environment, where UK government policy and funders' requirements more strongly emphasise value for impact from research, were found to guide the behaviour of scientists in the food research institute. A quote from one of the scientists illustrates this:

> *"Ten years ago, the word impact didn't exist, whereas now it's all they talk about and every single project has to have an impact statement, a pathway to impact"* (Scientist, BACCHUS Project)

Whilst previous literature views an incentive as a motivational construct and a tangible resource (such as bonuses and promotions), predominantly from an economic view [46,47], this study finds that nonpecuniary incentives are an important intangible resource for scientists. This result is congruent with Iorio et al. [22] view on the mission motivation

view of knowledge transfer. A nonpecuniary incentive mechanism is driven by explicit and implicit norms of compliance rather than from an economic perspective, and individuals then take action from these norms, which in this case reinforces researchers' behaviour to transfer their scientific knowledge. The issue of compliance sits at the management level and its wider environment, and not at the level of individual scientists. Therefore, it is a structural and not an individual mechanism.

Theorists have explored different approaches for explaining the complex factors and mechanisms that determine compliance. There are approaches that borrow insights from neo-institutionalist literature [48] whilst others develop complex models of individual rationality by borrowing concepts from psychology [49]. Another approach to compliance theories comes from the perspective of social norms [50] and this is the closest theoretical perspective to the incentive mechanism in this project. In this approach, social norms are understood in the sense of unwritten rules shared by a group, which are sanctioned, both positively and negatively, by the group's members.

### 4.1.2. Reputation

Reputation is identified as a key mechanism for knowledge transfer and highlights the importance of delivering results and maintaining good relationships with a wide range of actors. In this case study, reputation comes from past performance and perceived know-how; rare expertise; an international standing and specialised laboratory facilities; and established networks with industrial partners, as illustrated below:

> *"Clostridium botulinum is so difficult and dangerous to work with and only very few places in the world can do this sort of work and we are one of them. Industry wants to extend the shelf-life of their products and decrease the heat treatment, without adding preservatives. To do that they have to come to a lab with a series of expertise. Globally we are one of the few labs to do that . . . we are well known, we publish a lot and speak at a lot of conferences"* (Scientist, SUSSLE Project)

Reputation is based on opinion, or how much respect or admiration someone or something receives, based on past behaviour or character [51]. Lucas and Ogilvie [47] argue that reputation has a strong positive association with knowledge transfer. Coming from a perspective that knowledge transfer is important for the competitive advantage of firms, they look for factors that help to explain successful knowledge transfer. Reputation involves assumptions about the value of prior actions to future expectations. They conclude that good reputation facilitates knowledge transfer by reducing the need for constant monitoring between sender–receiver, which in turn improves transparency and speed of information sharing. Studies have also viewed reputation as a tool that organisations use to assess potential partners and reduce the inherent uncertainty within knowledge transfer relationships [52,53].

Theoretically, reputation can be explained using a trust theory lens. Trust theory is based primarily upon expectations of reciprocity or perceived utility in strategic interactions [54,55]. This rational perspective on trust is commonly based on predictability and past performance with relation to the costs and benefits of the action under consideration. The development of trust requires enough information for the trustor to make an assessment of expected outcomes. As such, trust is primarily cognitively based, consisting of the perceptions of the ability and integrity of the trustee combined with consistent past performance. The presence of trust enables knowledge sharing and knowledge transfer.

### 4.1.3. Opportunity

Opportunity is a mechanism which has a neutral connotation in this study, as scientists use their resources both to their own benefit regarding the pathway to impact, but also to attend to an industry need regarding compliance to a new regulation, as illustrated below:

> *"We (Food Databanks) as a group because we are a national capability, we have to be seen to be helping nationally this cause, the whole impact agenda . . . we have the sort of public engagement about everything we do . . . "* (Scientist, NIS Project)

> *"We produce these composition of foods, 3000 foods, and all the nutrients' composition. It's like a bible of what is in the food we eat, and because we produce it, we have to find other things to do with it. The new regulation coming in end of 2016 says all producers of food need to have a nutrition label on their food"* (Scientist, NIS Project)

Opportunity or opportunistic behaviour can be viewed negatively [56–59]. For example, Bouncken [60] study on alliances and open innovation in biotechnology firms suggests that opportunity is a negative mechanism that can be limited if firms employ specialised, complex and hidden knowledge, which hampers understanding by other firms. Thus, opportunistic behaviour among alliance partners could lead to mistrust and diminishing openness. Conner and Prahalad [61] however, argue that knowledge-based considerations outweigh opportunity. For example, they argue that individuals coordinate their productivity from knowledge-based resources rather than from just an economic, transaction cost perspective. Resources such as value and rarity can be valuable assets that also generate competitive advantage.

The opportunity mechanism in this research is not purely related to self-interest or a transaction cost perspective, but also from a utilitarian view of compliance theory. This is likely to arise as (1) the outcome of the project was fairly certain, i.e., there was a real need from industry which made the success of the project more likely; (2) there were already established networks that provided visibility to the service the project was offering; (3) the scientists have an established reputation in food composition and analysis. Thus, the theory advanced here does not dispense with economic arguments, but rather adds another layer to it from a commitment and compliance point of view.

Opportunity is the "conscious policy or practice of taking advantage of circumstances" [62]. External regulation offered the opportunity to fulfil their National Capability duties regarding pathways to impact. By taking advantage of their datasets, using current scientists and hiring a business development manager, they were able to strategically use this occasion to their advantage. Opportunity is traditionally explained from a transaction cost economics and self-interest perspective, "with guile" as coined by Williamson [63] (p. 255), where the lack of relational trust could lead individuals to break rules if they consider the utility of that violation exceeds the utility of being caught. Williamson [64] later elaborated the concept of opportunity in terms of "the incomplete or distorted disclosure of information, especially to calculated efforts to mislead, distort, disguise, obfuscate, or otherwise confuse" [64] (p. 47). His explanatory concentration on opportunity potentially ignores additional sources of organisational constraints and opportunities.

Unlike for the BACCHUS project, where an incentive mechanism was explained by compliance that originates from social norms, in NIS the opportunity mechanism can be explained from a utilitarian view of compliance. Nielsen and Mathiesen's [65] study on Danish fisheries presents an opportunistic approach to compliance on legislation. They argued that opportunistic behaviour influences compliance. However, opportunity does not equal violation or continuously breaking the regulations to obtain an economic gain. The high compliance in the Danish fisheries was primarily due to good legal financial opportunities, which consequently resulted in higher profitability for the fisherman. Similar to this example, it is argued that a compliance angle on the opportunity mechanism [65] is more in line with this NIS case than a transaction cost perspective [63]. This could be because (1) the outcome was fairly certain, i.e., there was a real need from industry which made the success of the project more likely; (2) there were already established networks that provided visibility to the service the project was offering; (3) the scientists have a fairly established reputation in food composition and analysis. Thus, the theory advanced here does not dispense with economic arguments, but rather adds another layer to it from a commitment and compliance point of view.

### 4.2. Action-Formation Mechanisms

The data analysis showed three dominant action-formation mechanisms: instrumental rationality, self-interest and strategic calculation. The action-formation mechanism seeks to explain the actor's behaviour and choice, looking for an explanation of why people act given their motives and situations.

#### 4.2.1. Instrumental Rationality

Some scientists adopt an instrumental stance to knowledge transfer, for example, to enable them to do more research, to be nearer the market or to access technologies and products that they do not have. This behaviour can be identified in the following quote:

*"We don't put products on the shelves, the food industry does. There's got to be an outcome to the research rather than just me getting some research papers. Ultimately, I can spend 40 years doing research in the lab and if it makes no difference to any product on a shelf and a consumer never gets a choice to buy something new, or at least has more certainty that it has a health benefit"* (Scientist, BACCHUS Project)

Instrumental rationality is "determined by expectations as to the behaviour of objects in the environment and of other human beings; these expectations are used as conditions or means for the attainment of the actor's own pursued end" [66] (p. 24). In other words, it concerns practical reasoning that helps one decide how to do things, in this case, how to do more research, how to access technologies and achieve more tangible results. Theoretically, instrumental rationality can be explained by several microtheories of rationality. Weber [66] was the first sociologist to distinguish two types of rationality that explain reasons for individuals to act and to believe. One type is instrumental rationality which represents acting efficiently to satisfy practical needs, whereas the other type is value rationality which represents acting to conform to impersonal social rules.

One theory that corroborates this context and helps to explain instrumental rationality as a mechanism is the theory of reasoned action (TRA). TRA assumes that human beings are usually rational and make systematic use of the information available to them [67]. According to TRA, an individual's behaviour is determined by their intention to perform that behaviour. Consequently, their intention is determined by their attitudes, subjective norms, and salient beliefs about the results or outcomes from their actions. Within this research, scientists' belief that they could add value to society was one aspect that determined their motivation and action to apply their research closer to the market and consumers.

#### 4.2.2. Self-Interest

Self-interest, defined as "the pursuit of personal advantage, be it money, fame, power, reputation, salvation" [68] (p. 68) is another key mechanism for science–industry relationships. This self-interest mechanism is exposed by an intrinsic motivation, where the personal benefit arises from the satisfaction of the end product. This can be exemplified by the following quote:

*"When you can safely extend the shelf life it has a massive effect not just on the profits of the company, but the reduction of waste, environmental concerns about reduced processing and so on. And all of those things are massive and possibly underrated . . . about a third of all the food manufactured is actually wasted but only a small part is from the consumer. A lot of it is from the manufacturing domain because of the way they have to satisfy certain regulations in terms of intermediate storage and so on. Understanding that leads to massive reductions in waste and energy usage and reduction in greenhouse gases, and all things like that, which most people don't see"* (Scientist, SUSSLE Project)

In this case, providing safer and less wasteful food products and the existence of indirect reciprocity suggest that scientists behave in a self-interested way to develop their reputation. The motivation to have an impact on wider societal issues and to contribute in a concrete way towards these challenges is a driver to participate in collaborative research. This mechanism has a consequentialist motivation, in other words, it is oriented towards

an outcome of action. In this case study, the outcome is safer and less wasteful food manufacturing processes. While the actions are observable—reaching end consumers, fulfilling an industry need, and so on—the motives are not observable. Therefore, the mechanism refers to the motive behind those actions, in this case the motivation to fulfil their personal and professional interests.

Drawing upon self-determination theory [69,70], self-interest can be defined as a value-driven mechanism. According to self-determination theory, individuals are motivated to act when they believe their behaviours will lead to desired outcomes. Self-determination has roots in social psychological needs and motivations. Niemiec and Ryan [71] explain educational practice from self-determination theory. They suggest that both intrinsic motivation and autonomous types of extrinsic motivation are conducive to engagement and optimal learning in educational contexts. Additionally, they suggest that supporting students' basic psychological needs for autonomy, competence and relatedness facilitates their autonomous self-regulation for learning, academic performance and wellbeing.

Autonomy, competence and relatedness were also identified within this research. For example, scientists showed an intrinsic motivation to tackle societal challenges and produce meaningful research, which represents the importance of relatedness. Such behaviours are internally perceived, which means they are experienced as emanating from the self rather than from external sources, indicating autonomy. Scientists were also extrinsically motivated by having the competence to retain research staff within their group and to fulfil an industry need. It can be argued that these were enacted to satisfy contingencies, such as the avoidance of ending up with a smaller research group or lacking the recognition of attending to an industry need.

### 4.2.3. Strategic Calculation

Strategic calculation relates to a general plan that is created to achieve a goal and is an approach that assumes a process characterised by exogenous, self-interested preferences and instrumentality [72]. In this case, scientists used a new regulation to exploit commercial opportunities within their National Capability, the Food Databanks. Having the ownership of the datasets and the skill sets to provide this service to food producer SMEs promoted a belief that they are offering an innovative and disruptive service. This belief was fuelled by their own motivation to help SMEs and for being seen as philanthropic, which is an exogenous way of being perceived. There is also an assumption driving the project that most SMEs find food compositional analysis prohibitively expensive. All of these assumptions and beliefs have motivated the project team to take action to create exposure and promote their innovative service.

> *"The project idea came about for two reasons, one we were sort of looking for opportunities to exploit some of the knowledge and data that we have. That was sort of driven by the national capability and what it should be doing. Then there is the need in terms of labelling. With the regulations changing, we knew that SMEs were in need of help"* (Scientist, NIS Project)

There was a strong focus on commercialisation, generating business opportunities and financial motivation for this project. Haeussler and Colyvas [73] argue that scientists accrue rewards through the scientific, human, and social capital from which they can draw for their work. This study supports this argument. Scientists used their scientific capital, i.e., databases, to create a service for SMEs, as well as their human capital to analyse recipe ingredients. Social capital also played a key part in this project, as it was through the scientists' networks that the product was made visible to SMEs. Thus, social ties with industry created a positive association with the commercial engagement.

Strategic calculation assumes that individuals are intentional actors. Theoretically, rational choice [45,66,74,75], a branch of game theory, deals with the relations and actions socially committed among rational agents, offering a good explanation for the mechanism of strategic calculation. Game theory is a branch of applied mathematics used as a theory to explain the rational side of social science. Being a microlevel theory, rational choice assumes

that individuals are the basic agents of social phenomena and that their rationality is the causal mechanism that produces events in the social world. It assumes that individuals are purposive, goal-oriented, and intentional actors. Beyond this, however, rational choice theory does not directly identify the content of any individual interests or choice options. In this case, it can be argued that scientists' actions were intentionally geared toward the commercialisation of this service, with a clear goal of financial impact.

Although rational choice theory offers a good explanation for the mechanism of strategic calculation regarding scientists' motivations, it offers a weaker explanation regarding scientists' beliefs. A softer version of rational choice theory, such as Boudon's [76] Cognitivist Model could explain this further. The Cognitivist Model supposes that actions and beliefs are "meaningful to the actor in the sense that they are perceived as grounded on reasons" [76] (p.191). It can be argued that scientists' belief that they were offering an innovative service and consequently helping SMEs is confirmed through their networks and reasoning that small manufacturers would struggle to afford compositional analysis.

### 4.3. Transformational Mechanisms

The data analysis reveals three key transformational mechanisms: aggregation, learning and adaptive self-regulation. These transformational mechanisms occur when individuals, through their actions and interactions, generate various intended and unintended social outcomes.

### 4.3.1. Aggregation

The aggregation mechanism denotes "any process in which actors who may have initially different preferences interact to bring about a decision that all of them accept as binding" [68] (p. 400). It involves different decision-making styles and methods. Many examples given by the participants involved some level of arguing or voting, such as when a decision had to be made on the use of human studies within a project or the decision to recruit a new SME to the project because the previous one had gone bankrupt.

There are some studies that look at decision-making and communication aspects in the knowledge transfer literature [11,13,77,78]. For example, Chung-Jen et al. [13] explored how cooperative competency, which includes trust, communication and coordination, has a mediating role between transfer mechanisms such as replication and adaptation, and how these affect knowledge transfer performance. There is a difference between this study and Chung-Jen et al. [13] study in relation to mechanisms. The latter views mechanisms of replication and adaptation as the process by which firms receive knowledge. Consequently, this view of mechanism is the process itself and how firms change their operations due to new knowledge. This study however views mechanisms as generative elements, considering instead which mechanisms lead to replication or adaptation.

On the other hand, there are similarities between this study and Chung-Jen et al. [13] on the communication and coordination aspects, which are considered to be critical elements for successful knowledge transfer. Communication includes formal and informal sharing of information, and coordination refers to how activities, people, routines and assignments work together to achieve a goal. Chung-Jen et al. [13] finds that increased communication, through shared language and symbols, and more effective coordination to use the sender's knowledge in the recipient's context, help to increase knowledge transfer performance. This is partly in agreement with this study, where communication and coordination efforts played a key role in the success of the project. For instance, partners used meetings and interactions to understand each other's responsibilities, and giving and receiving feedback played an important role in moving the project forward, as illustrated in the following quote:

> "By telling everybody what you've done, people can give feedback, question things, suggest ways to do it better. If you think, actually if you change that, question why you are doing it differently, even if it doesn't say in the contract . . . we can go back and change the contract" (Scientist, BACCHUS Project)

Theoretically, aggregation can be explained by decision theory with perspectives from organisational procedures [79] and to a certain extent from political views [80]. March [79] contributed to decision theory with an organisational procedures perspective, which seeks to understand decisions as the output of standard operating procedures invoked by its subunits. This theoretical angle helps to explain aggregation in various ways: the formal standard procedure offered by the case study project funder provided specific guidelines for certain decisions. This method worked up to a point, but further decisions arose in the project that were not covered by the funder's guidelines. Therefore, the partners, which included industry and science, had to come up with methods that would work for all. Being a fairly large group, meetings had to be tightly structured and bargaining succinct.

The political view on decision-making [80] sees it as a personalised bargaining process, driven by the agendas of participants rather than by rational processes. Individuals differ on goals, values and the relevance of information. This political view can partly explain aggregation, particularly when it refers to bargaining. The decision-making context was one of the main contributors to the project's success. It can be argued that one of the reasons for this was that the partners adopted various methods to negotiate and adapt to each other's value systems. Another aspect that facilitated decision-making was that partners seem to have kept the SMEs' frame of reference throughout the project, which provided a common focus. It can be argued, therefore, that it helps when facilitating decision-making in knowledge transfer relationships to keep the context and the customers' frame of mind central.

### 4.3.2. Learning

Learning is a process where individuals absorb something new such as knowledge, a skill, behaviour or value that they did not know before [81]. The learning mechanism was rooted in two ways: (1) project partners learned from each other during meetings and discussions, and (2) food manufacturer SMEs learned from tools and guidance provided by the research institute during the training workshops, as illustrated below:

> *"We have a series of technical training sessions for CFA members that are still going on. Some of them here (IFR) and others in Kettering. So Mike and I give an afternoon where we talk about how the project went and what the results were. I demonstrate some software tools that we generated during the project and how they can use them in their business"* (Scientist, SUSSLE Project)

> *"We've been running implementation workshops together: what the activities were, what the findings were and how to implement them. I drafted the implementation guidance"* (Scientist, SUSSLE Project)

The outcome intended during these actions and interactions is to translate the findings of the project into a digestible and useful tool for food manufacturers. From applying these novel processing methods, food manufacturers were able to produce safer and less wasteful products, which are now on supermarket shelves. It can also be argued that partners learned from each other during the project. Although partners had a similar final goal, they came from different industries—food manufacturing, academia and science—and had different ways of working, which meant they had to adapt and learn new ways of making collective decisions within the project.

Bercovitz and Feldman [14] argue that academics' decision to disclose their findings appears to be influenced by peer effect, where learning activity occurs within a cohort of peers with similar characteristics. This view supports the findings from this research, and even though there were partners from industry, academia and science, they all had a goal in common and shared an expertise in the subject matter.

Levine et al. [82] examine how group members develop a shared reality through their interaction with one another and how that shared reality shapes their problem-solving and learning strategies. This view reflects some of the examples from this study, such as the interactions among partners during meetings and discussions and how, during the project,

these interactions created a sense of group identity which promoted knowledge sharing and the ability to give and receive feedback freely. Following this aspect of interactions, Muthusamy and White [6] argue that social interactions and exchanges between partners are imperative for knowledge transfer success, which facilitates organisational learning. They also argue that the greater the reciprocal commitment, the greater the degree of learning accomplished. The aspect of social interactions came across in this study. It can be argued that because partners were from different countries, they perhaps did not have many opportunities to interact socially, although they did meet fairly often. As they met in a country or place of business other than their own, it could be argued that this foreign environment would make social interactions even closer. Moreover, what came across was the sense of belonging to a team, which in turn facilitated discussions and learning

A different view on learning is offered by Braun and Benninghoff [83]. They look at rationality in the learning processes of research policies. They concluded that learning processes are a mix of rational and nonrational elements and that all learning processes may have a combination of interest and power. Although this study supports the claim regarding interest, it does not support the claim regarding power, as this was not evident in the findings. It could be argued that power did not come across strongly because this was a project environment where partners saw each other more equally, whilst for Braun and Benninghoff [83] study, the context was learning in a policy environment, where conflict and power are more visible.

### 4.3.3. Adaptive Self-Regulation

Adaptive self-regulation can be defined as when individuals "can respond to the complexity and dynamic pace of their immediate environment in a timely fashion" [84] (p. 93). In this case, actors responded to critical processes such as the decision to create the strategic alliance very quickly. This could be explained in that the legislation was fast approaching, therefore they needed to get the service to food producers in a timely manner. Thus, it can be argued that decisions were driven by an adaptive self-regulation mechanism, as individuals involved in the project had to make autonomous and quick decisions, even though they were restricted within a social structure consisting of multiple constituencies or stakeholders, as illustrated below:

> "I've made the decisions alone on how to go about meeting SMEs and doing all the marketing. It was very empowering because they felt confident I could do that, they believed I had the experience. We do work really well as a team because I need them, they need me" (Scientist, NIS Project)

This kind of mechanism tends to have a functionalist explanation, where the consequences of the actions for a certain situation have a purpose and will produce a beneficial effect. In this case, for example, an actor intentionally looked for alternatives to grow and sustain the project, deciding on a strategic alliance with a Statistical Analysis Software (SAS) firm. This particular action was created with the purpose of future financial sustainability.

There are not many studies that look at adaptive self-regulation in the knowledge transfer literature, however there are studies that look into elements from the narrative that led to this mechanism such as autonomy, teamwork and empowerment. Molina and Llorens-Montes [85] consider how teamwork and an increase in individuals' autonomy affect knowledge transfer. They concluded that teamwork improves knowledge transfer, however greater autonomy only increases knowledge transfer when there are difficulties such as high tacitness. The findings from this case support this view of autonomy, teamwork and tacit knowledge. For example, the project members worked as a self-directing team and the roles of scientists, the business manager and SMEs were well-defined. The tacitness of the service they provided to SMEs, although it was straightforward to scientists, was something that SMEs could not have calculated by themselves, with the alternative being to have their food products analysed in laboratories rather than via recipes.

Following a similar context around autonomy and knowledge transfer, Llopis and Foss [86] tested a model of intrinsic motivation and autonomy as moderators of knowledge

transfer relationships. They suggest that an environment that emphasises efforts towards groups, rather than individual outcomes, is better for knowledge sharing when individuals show low levels of intrinsic motivation, but high levels of autonomy. This view is also shared by Ozlati [7], who suggests that organisations can increase knowledge sharing by encouraging individuals' autonomy. Although this study supports the view on autonomy, it does not entirely support the findings related to intrinsic motivations. Scientists were predominantly extrinsically motivated to undertake this project in order to further business opportunities and use their databases for impact. However, there was a motivation to be seen as philanthropic, which can be argued to be intrinsically rooted.

Another element that relates to individuals' interactions is their ability to organise themselves independently. Studies such as Jobidon et al. [87] refer to the relevance of self-organising teams and role variability, arguing that high variability of individual's roles within teams is associated with poorer performance and coordination. They concluded that individual's role flexibility can be beneficial, however, high role variability can cause ambiguity and consequently negatively affects goal achievement. This view aligns well with the findings in this case. For example, individuals had well-defined roles and they also had a lot of flexibility to make decisions independently and to interact freely with each other. Consequently, the variability of roles was low, which could be argued to provide an effective way for the team to work together.

Tu et al. [88] explore the process through which a team dismantles its existing order and rebuilds a new one via innovations and changes which are spontaneously initiated by team members. This concurs with findings from this research, where project members came together and had to find a new way of executing their roles to deliver well-defined tasks and to identify and deliver new tasks. For example, scientists had built the databases, and they had to develop specific software to extract the kind of information needed to match the new service they were offering to SMEs. Another element congruent with this study is in relation to feedback structures. One of the reasons the team worked well independently is because they had an effective feedback loop during meetings and discussions.

Although these studies offer detailed processes for self-organising teams and for the impact of autonomy and the relevance of teamwork in knowledge transfer relationships, they do not offer explanations based on generative mechanisms. A theoretical explanation to adaptive self-regulation can be offered by empowerment theory [89]. Empowerment theory often refers to processes of giving individuals greater discretion and resources, to increase their degree of autonomy and self-determination to act on their own authority. This distribution of power helps individuals to take control of their circumstances and achieve goals.

This theoretical angle helps to explain adaptive self-regulation in this case as: (1) all project members had a high degree of discretion regarding resources and decision-making; (2) the business development manager had authority to make decisions independently; (3) there was a strong culture of trust and members were confident with their responsibilities. It can be argued that an appropriate structure and information and communication system was in place, with meetings and discussions only scheduled when critical decisions or approvals were needed:

> *"We engage and we discuss. The scientists trust my business experience and I trust their scientific knowledge. I presented strategically the issues that I saw with NIS that was going to be good for small businesses"* (Project Manager, NIS Project)

This system encouraged individuals to act independently and in a self-motivated fashion. The boundaries and well-defined tasks created autonomy and efficient decision making.

## 5. Discussion

In much of the literature there is a proliferation of macro level constructs which can be problematic because the micro mechanisms that influence knowledge transfer and its outcomes are seldom identified. By grounding the knowledge transfer debate in a frame-

work that bridges macro and micro levels, this study contributes to the emerging body of literature on mechanismic explanations [90–92]. The focus on generative mechanisms in multiple projects marks an advance over earlier methodologies and theorising. Rather than employing vague notions of correlations between variables, the theorisation has been based on generative knowledge transfer mechanisms. The combination of micro and macro perspectives complements and adds to the knowledge transfer literature by delving into a deeper ontological layer. In addition, each mechanism is explained based on further theoretical perspectives and how these influence practice. By unpacking the microfoundations, the interactions and the macro influence on each knowledge transfer project, this study provided an in-depth mechanismic explanation that adds to previous research.

At a macro level, structural conditions influenced each of the projects differently. Whilst incentive has been viewed in the literature as either carrot or stick [46,47] and as a tangible resource [15,21] predominantly from an economic angle (better pay, promotion, bonuses), this study positions nonpecuniary incentive as a generative mechanism driven by social norms and social cohesion. The implications of this view are that a focus on social relationships and interpersonal interactions are more important than financial rewards, which is an outlook shared with studies on how organisational climate affects subjective norms [93]. Similarly, reputation is a generative mechanism which is directly related to the international reputation of the research institute. This mechanism has influenced the application of the project as a driving or enabling mechanism, which differs from current literature that sees reputation at an individual level as a motivation to engage in knowledge transfer activities [20]. This view of reputation as an enabler has direct implications regarding the perceived importance of institutes or organisations as a whole rather than the reputation of individuals. Opportunity is a mechanism that was driven by a new national regulation that food manufactures had to comply to. This is not a mechanism explored in current literature, however, opportunistic elements such as having the right facilitating conditions [94] has been seen as an opportunity to engage in knowledge transfer.

The micro generative mechanisms are closely linked to scientists' intrinsic motivations to engage in knowledge transfer and often play a prominent role to drive knowledge transfer activities. It is argued that scientists' entrepreneurial commitments are driven by rational and relational-type generative mechanisms which are rooted in individual's motivations and beliefs and can be explained by different micro theories. Instrumental rationality is derived from the willingness to be nearer the market and to access technologies and products from practitioners, whilst self-interest comes from the motivation to have an impact on wider societal issues and strategic calculation comes from the exploitation of a commercial opportunity. This resonates with studies such as Lam's [21] "puzzle", which refer to the satisfaction derived from puzzle-solving activities but also from contributing to the knowledge of society and from prosocial behaviours such as mission [22]. Similarly, Ramos-Vielba et al. [95] find that intrinsic motivations are important in their analysis of the motivations and barriers to scientific research groups' cooperation with firms and government agencies in Spain. They derive three categories of motivations: advancing research, applying knowledge and accessing financial resources.

Transformational mechanisms show the interactions among individuals and have several implications. Aggregation meant that individuals had to adapt to each other's styles of decision-making, which means persuasion skills and sensitivity to others' value systems are important implications for management. Learning means a sense of belonging and group identity are important, whilst adaptive self-regulation means that autonomy and decentralised control facilitate knowledge transfer. The implications for these kinds of independent interactions are that communication channels should be transparent and well-defined roles and tasks help with clarity and effective execution.

There is a direct implication for organisational rules and policies. By understanding the generative mechanisms that drive knowledge transfer, it is possible to design organisational rules and policies that are more effective. Identifying these mechanisms not only provides evidence for policies, but also distinguishes generic factors from those that arise from

unique projects. This, in turn, should lead to more informed contributions to academic–industry relationships and, arguably, to more effective support for knowledge transfer between scientists and industry.

Using this approach, this research finds that a construction of relationships based around social norms, autonomy and relatedness are more optimal in science–industry knowledge transfer relationships than a focus on financial incentives or transaction cost theories [62]. Scientists draw from social norms and act in an instrumental way to solve problems. Considering how organisational and national policies support or undermine the norms of self-interest and nonpecuniary incentives could offer more satisfactory knowledge transfer results. A reliance on solely improving access to funding is likely to be of limited effectiveness in increasing science–industry engagement, whereas an increased emphasis on tackling research compatibility may be more fruitful.

Reputation as a mechanism for knowledge transfer reinforces the idea that the process is highly dependent on the relevance and quality of research that scientists develop and also reflects the importance of trust in social interactions and the strong influence of the relationships that scientists establish within and outside organisational boundaries. One of the implications for reputation as a mechanism for knowledge transfer is that it carries a visible perceived status. In fact, reputation is part of the class of intangible assets identified as social approval assets, because they derive their value from favourable collective perceptions.

Public policy often seeks evidence-based research findings. Typically, researchers carry out experiments or surveys. Although these studies provide useful outcomes, they do not identify the mechanisms that explain the outcomes. Identifying the mechanisms, whilst also distinguishing generic factors, should lead to more informed contributions to public policy making and, arguably, to more effective support for knowledge transfer between scientists and industry. For example, the UK Research Councils assess researcher progress and performance. One of the items within their assessment relates to the research organisations' achievements in knowledge exchange and commercialisation (KEC). KEC has a strong focus on direct financial impact through commercialisation and support in economic competitiveness. From the findings in this study, a strong focus on economic competitiveness might work against KEC activities unless nonpecuniary incentives such as relevance, morality and status are also included. Other implications for policy are related to strategies that enhance scientists' autonomy, relatedness and competence, which would offer better outcomes. Thus, providing choices of projects with meaningful rationales for the application of their science could improve knowledge transfer. Another key aspect for autonomy is the minimisation of control. It can be argued that an organisational environment that focuses on applied science could enhance the perceived relatedness need. Furthermore, strategies to enhance competence could involve subject familiarity and exposure to industrial communities.

## 6. Conclusions

This study represents the first attempt to systematically analyse, from a mechanismic perspective, how knowledge is transferred from science to industry. This mechanismic view integrates a multilayered framework which offers direct implications for organisational policy. Whilst the traditional view of scientists as producers of scientific discoveries is outdated, there is a reluctance to see them as pure commercialisers, who pursue commercial activities mainly to obtain the much needed funding for research in an increasingly resource constrained environment. Evidence based on the interviews suggests that the position and motivations of scientists are not fully determined by their commercial orientations, but have nonpecuniary and socially related drivers which influence their efforts.

Drawing on social psychology theories, this study offers important insights into the social and personal mechanisms driving the knowledge transfer and the commercialisation behaviour of scientists. These mechanisms have been recognised by social psychologists as a pervasive and powerful driver of human action, but they are neglected in much of

the existing research on academic entrepreneurship. This study suggests that a fuller explanation of scientists' commercial behaviour will need to consider a broader mix of mechanisms that goes beyond economics and extrinsic rewards to include nonpecuniary, social and personal aspects related to intrinsic motivation.

Fostering an incentive that gives value to sharing behaviours is likely to increase the mutual social exchange relationships that are apparently important in driving knowledge transfer intentions. It can also be argued that providing a work environment characterised by high levels of organisational citizenship would support the formation of robust communities within research organisations, consequently supporting the social norms of sharing.

Knowledge transfer is a socially situated activity, therefore individuals' motivations and beliefs (action-formation mechanisms), interactions (transformational mechanisms) and their environments (situational mechanisms) are important elements in understanding this process. The importance of the social context helps to explain why individuals get involved in knowledge transfer. Mechanisms such as nonpecuniary incentives, self-interest and strategic calculation show that individuals engage in knowledge transfer if there are social norms in place, if they are sending or acquiring knowledge from similarly reputable partners, and are operating in a culture that encourages sharing. It can be argued that these mechanisms help develop a sense of ownership whereby scientists feel a personal affinity to the knowledge transfer process effort and are committed to its success. Scientists' personal interest in knowledge application also appears to strengthen a strong professional conviction to make their knowledge socially relevant. Therefore, an organisational environment conducive of transparency with a focus on scientists' belief systems is more like to be successful than a focus simply on industry engagement.

**Author Contributions:** Conceptualization, K.Z.-L.; methodology, K.Z.-L. and F.L.; software, K.Z.-L.; validation, K.Z.-L. and F.L.; formal analysis, K.Z.-L. and F.L.; investigation, K.Z.-L.; resources, K.Z.-L.; data curation, K.Z.-L.; writing—original draft preparation, K.Z.-L. and F.L.; writing—review and editing, K.Z.-L. and F.L.; visualization, K.Z.-L.; supervision, K.Z.-L. and F.L.; project administration, K.Z.-L. All authors have read and agreed to the published version of the manuscript.

**Funding:** This research received no external funding.

**Data Availability Statement:** The anonymised data presented in this study are available on request from the corresponding author.

**Conflicts of Interest:** The authors declare no conflict of interest.

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
