# Peer review of "Generative Mechanisms for Scientific Knowledge Transfer in the Food Industry"

_sustainability, doi:10.3390/su13020955_

Round 1
Reviewer 1 Report
Thanks for the article, I enjoyed reading it. The authors analyze the mechanisms for scientific knowledge transfer in the food industry. Although the topic is very interesting and necessary, there are significant aspects in the manuscript that I believe can be improved to make a significant contribution. I hope my comments will help you:
- I think it would be interesting to analyze how the transfer is materialized, through the constitution of spin-offs, formalization of contracts, establishment of royalties, etc. It would also be interesting to see the importance of each type.
When analyzing the transference, little importance is given to the inconveniences, such as:
- I think it would be necessary to develop more and to reference the difficulty of understanding between the food sector and researchers. In many occasions both agents pursue different objectives.
- I don't see any reference to the cost of knowledge transfer for the SME. Researchers are concerned about the cost-benefit ratio for SME.
- As you can see in the article, among the objectives of the transfer is to create wealth and make money. It would be interesting to see among the companies that apply the knowledge which is the evolution of their financial indicators, such as: sales, EBITDA, profitability.
Hope these comments help to develop such a potential work.
Author Response
Dear Reviewer,
We thank you for your supportive comments on our work; these comments have helped us to improve the manuscript. Please find attached our response specific comments and track changes within the manuscript.
Best wishes,
Karla

Reviewer 2 Report
I find potential in this research, and I believe that you are exploring interesting and relevant questions that could potentially contribute to the literature of knowledge management. I think the paper is promising because it tries to provide a richer and more composite understanding of this topic.
However, from my point of view, some improvements are needed, which I highlight below.
- General
- You are studying a unique context which you could take more advantage of. In your introduction as well as in the results section you could strengthen your argumentation by referring consistently to the food industry.
- Introduction
- I think the introduction of your paper needs to be sharper and more convincing about the gap in the literature that you are trying to address as well as its relevance. In the current version, the introduction might not be attractive. From my perspective, the introduction of your study could be structured in a different way, highlighting the objective and the novelty aspects, which represent the contributions of your research. You need to convince the readers about the novelty and relevance of your study in the introduction. Moreover, you could include the contributions of your paper to the literature. In the current version, the reader does not know what the contributions of this study are.
- For example, I don’t understand the reference to the Food Research Institute in the second line of the introduction. I think that it paper’s objective is not the Food Research Institute. Similarly, I’m confused because the objective and contributions are in the last two paragraphs of the Literature Review section. I usually expect to find that information in the introduction section, whereas the literature review one has to focus on the theoretical framework of the paper.
- Theoretical framework or Literature review
- The paper seems to discuss the relevant conceptual concepts appropriately, and the paper’s references are up to date. However, the study needs to go more deeply into previous theoretical framework in order to improve the literature review. Particularly, you need to go more deeply into previous theoretical framework that considers the knowledge transfer in the context of academic-industry relations. In this sense, I feel that you have forgotten papers that analyse this relationship.
- Similarly, as you recognize, the reward system and motivation to transfer knowledge in academia are quite different than those of other types of organizations, so you need to make more efforts to review the previous literature in the context of scientific production and relationship with SME in academia.
- Methods
- I agree with the qualitative research that you consider in your paper, but I have some doubts. For example, in page 4, you poit out “[…] … utilizing a multiple framework, empirical data from fifty-two in-depth interviews with scientists and directors of food SMEs, seventeen supporting project documents, and sixteen observations were reviewed and analysed”. Maybe, you could explain what is the meaning of “sixteen observations”?
- Based on the previous paragraph, I would like to have information about the 52 in-depth interviews, who were the people interviewed? How many were scientists and how many were directors of SME? That is, I think that the paper would be improved if you provide more information about the characteristics of the interviewed individuals.
Other changes:
- You should review some sentences because they could be not proper.
I hope these comments are helpful to improve your paper.
Good luck
Author Response

(The authors gave the same response as above.)

Reviewer 3 Report
The paper deals with a very interesting topic and provides a good basis for interesting future research. The paper is original, written in good English, with appropriate methodology and theoretical background as well as a good literature review. The aim of the paper was clearly formulated. The structure of the paper is logical, text is easy to read. The findings are presented and discussed. The aim of the paper was to investigate the generative mechanisms for scientific knowledge transfer in the food industry, addressing the sustainability of knowledge transfer projects related to health, safety and regulation. It should be taken into account that the research still has some limitations (well described by authors) but there are a lot of potential directions for further research, so good luck! Congratulations to the authors for this interesting article.
Author Response

(The authors gave the same response as above.)

Round 2
Reviewer 2 Report
I think you did a good job in addressing my comments. I want to congratulate you on this nice paper.